# Effects of Modifying Agent and Conductive Hybrid Filler on Butyl Rubber Properties: Mechanical, Thermo-Mechanical, Dynamical and Re-Crosslinking Properties

**DOI:** 10.3390/polym15194023

**Published:** 2023-10-08

**Authors:** Piyawedee Luangchuang, Tanawat Sornanankul, Yeampon Nakaramontri

**Affiliations:** Sustainable Polymer & Innovative Composite Materials Research Group, Department of Chemistry, Faculty of Science, King Mongkut’s University of Technology Thonburi, Bangkok 10140, Thailand; piyawadee232800@gmail.com (P.L.); artists2248@gmail.com (T.S.)

**Keywords:** bromobutyl rubber, self-healing, re-crosslink, tire application

## Abstract

Ionic crosslinking of bromobutyl rubber (BIIR) composites was prepared using butylimidazole (IM) and ionic liquid (IL), combined with carbon nanotubes (CNT) and conductive carbon black (CCB) to enhance the intrinsic properties and heal ability of the resulting composites. Variation in the BIIR/CNT-CCB/IM/IL ratios was investigated to determine the appropriate formulation for healing the composites. Results showed that the mechanical properties were increased until the IM:IL:CNT/CCB ratio reached 1:1:1/1.5, corresponding to the optimal concentration of 5:5:5/7.5 phr. Thermo-oxidative degradation, as indicated using temperature scanning stress relaxation (TSSR), demonstrated the decomposition of the composites at higher temperatures, highlighting the superior resistance provided by the proper formulation of BIIR composites. Additionally, the conditions for the healing procedure were examined by applying pressure, temperature, and time. It was observed that the composites exhibited good elasticity at 0 °C and 60 °C, with a high rate of re-crosslinking achieved under appropriate pressure and temperature conditions. This research aims to develop a formulation suitable for the tire tread and inner liner of commercial car tires together with artificial skin products.

## 1. Introduction

In the field of polymer composite materials, particularly in rubber nanocomposites, the concept of elastically self-healing composites has recently garnered a lot of attention [1,2,3], both with and without additional fillers. It has been recognized that the healing ability of these composites operates by replacing covalent bonding with ionic crosslinking. As a result, using chemical and physical modification involving the insertion of cations and anions into the bulk matrix, crosslinking can be reorganized due to the attraction of oppositely charged ions. However, even though this discovery was made several years ago, it is not straightforward to use in specific applications. This is mainly due to the poor strength of composites after the ionic bonding, which is inherently brittle and prone to cracking. To achieve self-healing functionality and improve performance, cutting-edge additives such as ionic chemicals (e.g., butylimidazole (IM) and/or ionic liquids (IL) [4]) and hardened fillers (carbon nanotubes (CNT) [5,6] and conductive carbon black (CCB) [7,8]) have been applied to self-healing epoxidized natural rubber (ENR) and bromobutyl rubber (BIIR) composites. This approach is considered promising in terms of the mixing conditions and chemical ratios relative to the rubber concentration. In particular, in applications related to the automotive, aerospace, and pharmaceutical industries, BIIR—a butyl rubber derivative—is used due to its superior gas and moisture barrier properties. However, like all elastomers, it is susceptible to damage from mechanical stress, environmental factors, and other external influences. Therefore, incorporating self-healing mechanisms into BIIR composites has become a promising approach to extend their lifetime and enhance their performance under severe conditions [9,10].

In the past, with regard to introducing self-healing ability into composites, IM has been predominantly used due to its ease in performing the chemical modification process. IM, an organic compound [9,10], acts as a potential healing agent within a BIIR matrix. Before the material is damaged, it remains in a passive state. However, when damage occurs, it can trigger chemical reactions to create elastic links that can mend and restore the integrity of the composite materials. Since IM is used to create ionic bonds among the modified BIIR molecular chains, the extent of chain movement can be a key factor in facilitating the self-healing process. However, even though solely BIIR demonstrates effective re-crosslinking, the addition of fillers and hybrid fillers may have the opposite effect due to the absorption of rubber molecules onto the filler surface. Support from electricity and heat can resolve this issue owing to their superior electrical and thermal conductivities. However, it is evident that bulk composites require multifunctional lubricants to aid in chain movement during damage. Consequently, the incorporation of fillers, such as CNT and CCB, along with additional IL, poses a challenge in terms of determining the suitable ratio for IM, CNT/CCB, and IL within the BIIR composites, which has not been previously researched [11,12]. CNT and CCB offer excellent mechanical properties, including high tensile strength and aspect ratio, while carbon contributes to electrical conductivity and reinforcement. This synergistic combination of fillers optimizes the self-healing process and enhances the mechanical properties of the composite material [13,14]. Additionally, the inclusion of IL in the composite has improved its self-healing capacity. When dispersed in a BIIR matrix, IL promotes rapid flow and efficient healing responses when exposed to external stimuli [11]. The ionic nature of the liquid fosters electrostatic interactions in the composite, thereby enhancing its mechanical properties and healing efficiency [12].

Therefore, in the present work, the combination of IM, IL, and the hybrid filler, CNT:CCB, in BIIR composites, represents a groundbreaking advancement in self-healing materials. The objective of this study is to enhance the self-healing ability of BIIR composites while simultaneously improving their mechanical and thermo-mechanical properties by identifying the appropriate ratio of the additional chemicals. Furthermore, improved healing factors have been reported. The ultimate goal is to encourage the utilization of such composites in proposed tire and artificial skin applications.

## 2. Experimental and Characterization

### 2.1. Materials

Bromobutyl rubber (BIIR) was purchased from ExxonMobil Co., Ltd. (Houston, TX, USA). The multi-wall carbon nanotubes (CNT), grade NC7000, were manufactured from Nanocyl S.A. (Sambreville, Belgium). In addition, Carbon black (CB) and Vulcan XC72 were purchased from Cabot Corporation (Pampa, TX, USA). 1-butylimidazole (IM) and 1- butyl-3-methylimidazolium bromide (IL) were received from Merck KGaA (Darmstadt, Germany). Stearic acid was procured from Imperial Chemical Co., Ltd. (Pathum Thani, Thailand). Zinc oxide (ZnO) and sulfur were manufactured by Global Chemical Co., Ltd. (Samutprakarn, Thailand) and Ajax Chemical Co., Ltd. (Samutprakarn, Thailand), respectively. 2,2′-Dithiobis-(benzothiazole) (MBTS) was supplied by Flexsys Inc. (Ann Arbor, MI, USA)

### 2.2. Compound Preparations

The preparation procedure for the BIIR/ENR composite blends was carried out according to the formulation shown in Table 1. Initially, the modification of BIIR was performed using an internal mixer (Brabender VR GmbH and Co. KG, Duisburg, Germany) at 40 °C and a rotor speed of 60 rpm. In this step, BIIR was masticated for 2 min before adding the activators (i.e., stearic acid and ZnO), and mixing continued for an additional 2 min. Subsequently, the CNT-CB hybrid fillers were added, and mixing continued for another 6 min. IM and IL were then introduced into the rubber, and mixing continued for 4 min. Following that, the curatives (i.e., MBTS and sulfur) were consecutively added to the rubber compound, and mixing continued until a total mixing time of 16 min was achieved. The composite was sheeted at 160 °C using compression molding to obtain crosslinked composites with dimensions of 150 mm x 160 mm x 2 mm. It is worth noting that the modified BIIR, with the addition of agents (IM and IL) and the hybrid filler (CNT:CB), was designated using the code “M-BIIRx,” where “x” represents the modifying agent.

## 3. Characterization

### 3.1. Mechanical Properties 

Mechanical properties, based on tensile testing following ISO 527 (type 5A), were examined using a maximum of 5 specimens for each formulation. The tests were conducted at room temperature (23 ± 2 °C) with a crosshead speed of 200 mm/min using a tensile testing machine (Zwick Z 1545, Zwick GmbH and Co. KG, Ulm, Germany). To measure self-healing ability, dumbbell-shaped samples were also cut using a sharp razor blade (Energizer^®^ Holdings, Inc., St. Louis, MI, USA) before being pressed into the mold. The samples connected to the mold were heated at 120 °C for 30 min without applying any pressure. The healed samples were then removed and placed in a desiccator for 24 h, and their tensile properties were measured under the same conditions as the unhealed case. The healing efficiency (*X**), which is the ratio of the obtained values before (*X*′) and after (*X*″) healing conditioning, was calculated using Equation (1):(1)$$\;X^{*}\left(\%\right)=\frac{X^{\prime \prime }}{X^{\prime }}\;\times \;100$$

### 3.2. Morphologies

The surface morphologies of the composites, both filled and unfilled with TESPT, were elucidated using an optical microscope (OM) from Carl Zeiss Microscopy GmbH in Oberkochen, Germany. The samples were rapidly cut using a razor blade to obtain a smooth surface before measuring. This investigation aims to affirm the healing performance of the provided composites.

### 3.3. Abrasion Resistance

Abrasion testing of the composites was conducted to indicate their surface detachment resistance while performing in the Taber tester (Model GT-7012-T, GoTech, Taichung, Taiwan), using a maximum of 5 specimens for each formulation. The samples, approximately 110 mm in diameter, were polished under a 72 rpm rolling speed for 1000 cycles, and the wear index (*WI*) was calculated following Equation (2):*WI* (%) = (*W_y_* − *W_x_*) × 100/*W_y_*
(2)

### 3.4. Thermo-Mechanical Properties

To investigate the interaction between elastomer and filler mixtures, a Temperature Scanning Stress Relaxation Mechanism (TSSR) was employed using ISO 527 Type A standard samples installed within a TSSR testing chamber. Subsequently, an isothermal test was conducted by applying a 50% strain to the samples at 23 °C for a duration of 2 h as a function of temperature. It is noted that the relaxation spectrum (H(T)) was also calculated based on the relationship between relaxation modulus (E(T)) and temperature (T) using the following Equation (3):(3)$$H\left(T\right)=-T\;\left[\frac{dE\left(T\right)}{dT}\right]\;$$
where T is the temperature directly related to both E and H

### 3.5. Dynamic Mechanical Properties

The dynamic mechanical performance of the received composites was investigated using DMA 1 (Mettler-Toledo GmbH, Zurich, Switzerland). The tests were performed in tension mode with a frequency of 1 Hz and a strain of 0.2%. The properties were measured in terms of storage modulus, loss modulus, and loss factor (Tan δ) under a heating rate of 2 °C/min with temperature ranges from −90 to 80 °C.

### 3.6. Thermal Stability

Investigate the thermal stability characteristics of a mixed rubber by performing tests on molded samples using a Thermogravimetric Analyzer (TGA). Measurements were collected across a temperature spectrum ranging from 30 to 900 °C, with a heating rate set at 10 °C/min. Nitrogen (N_2_) was employed as the atmospheric environment from 30 to 600 °C, after which it transitioned to oxygen (O_2_) from 600 to 900 °C.

## 4. Results and Discussion

### 4.1. Tensile Properties and Morphologies

The mechanical efficiency of a composite is influenced by several factors, including the length of the molecular chains of the rubber, the reinforcement efficiency resulting from the specific surface area of the additional filler, and the interaction between the filler surfaces and rubber molecules [15]. Proper dispersion of the fillers within the selected rubber matrix leads to strong rubber-filler interactions, as re-agglomeration after compounding is hindered. On the other hand, if the filler-filler interaction is strong, agglomeration can occur, causing a higher number of defects within the matrix. Consequently, these defects are more susceptible to failure under external forces [16], which may lead to a reduction in the mechanical properties of the composites. Thus, according to the theoretical background of reinforcement, Figure 1 shows the tensile properties by means of tensile strength, elongation at break, and the 100% modulus of the BIIR composites with IM and IL modifying agents and hybrid fillers. Considering the composites without damage, there is a tendency for tensile strength and modulus at 100% elongation to increase compared to pure BIIR, and they increase with an increase in the IM/IL modified agent and hybrid filler. Because the alkylation between BIIR and IM interacts synergistically, the molecular chains are linked, and the rubber is strengthened, causing the reinforcing agent to adsorb onto the surface of the BIIR molecules. As a result, it enhances the ability to resist tension and increases the modulus. Increasing the modified agent of IM, IL, and hybrid fillers to 7 phr results in a decrease in tensile strength. This might be due to the molecular structure of the rubber; voids are formed because of the excessive presence of both IM and IL, caused by both linking molecules and extra filler molecules. Additionally, excessive reinforcement may lead to agglomeration. Consequently, the tensile strength decreases. Furthermore, considering the elongation at break, it is found that the elongation at break increases with increasing the modified agent of IM, IL, and hybrid fillers. The excess imidazole and ionic liquids present cause the molecular chains to move easily, increasing the elongation at break. However, the elongation at break slightly decreases after adding the modified agent of IM, IL, and hybrid fillers to 7 phr. Due to the adsorption of rubber molecules on the surface of the reinforcing agent, this effect can be greater than that of an excess crosslinking agent, resulting in reduced elongation at the break. On the other hand, concerning the composites after healing, although both tensile strength and 100% modulus tend to increase with the addition of the modified agent of IM, IL, and hybrid fillers to 7 phr, as also seen in Figure 1, the indicated values are lower than the cases without damage. This is, of course, dependent on the healing condition, which strongly relates to healing times. This can be seen in differences in tensile strength and modulus values before and after healing, where slightly different values are represented, especially in the case of 100% modulus. This means that, under controlled conditions, the damaged surface becomes healed, but it is not powerfully completed. The extension force in the initial stage at low strain, therefore, gives a higher degree of recovery than the tensile strength, which officially needs a high state of extension. However, comparing the received values of the composites after healing under modifying and filler variation, it was found that the same trends of increase in both properties with increasing such chemicals regarding the attraction of cations and anions to the BIIR molecular chains. This is also the rationale for the explanation of the elongation at the break of the composites, even though it becomes a constant situation due to excessive extension during testing conditions.

Self-healing efficiency is based on a combination of tensile strength, elongation at break, and modulus at 100% elongation of the composites. Figure 2 shows the healing efficiency of BIIR with and without the modified agent of IM/IL and hybrid fillers. This indicates that ion binding was well organized even though the hybrid filler was added. The curing of the composites was carried out in an oven at 120 °C for 30 min without pressure. This temperature allows for faster activation of ionic bonds for rebonding compared to unmodified composites. The high thermal conductivity of the cured properties of the added hybrid fillers causes cations and anions to move within the composite, creating molecular chains that can regenerate. Therefore, if the filler has good dispersion and is distributed throughout the composite matrix, healing can occur. In Figure 2, the healing efficiency provided by the module is higher than other mechanisms. This is due to ionic crosslinking, characterized by strong chemical bonds and high brittleness. Thus, at low strains, high stress in the composites was reported. The healing ability of composites containing BIIR with and without the modified agent of IM, IL, and hybrid fillers is confirmed by the optical microscope images displayed in Figure 3. The yellow line in the center of each sample is the bonded surface after 30 min of stress-free treatment at 120 °C. In Figure 3, it is found that on the surface of pure BIIR, there are marks of the cut that do not indicate molecular crosslinking. Although there are tight joint marks and no visible voids, the surface healing that occurs is a result of the rubber’s adhesive and soft properties when exposed to sufficient heat only [16]; it does not result from any bonding or strengthening within the molecular structure. Therefore, the molecular chains changing after the addition of the proper IM:IL:CNT/CCB ratios can be summarized in the proposed model of Figure 4 for the BIIR composites before and after modification by means of the molecular chains’ changes. It is seen that the modification of BIIR molecules with IM changes the BIIR chains to be ionic chains containing cationic and anionic charges, generating the recrosslinkable chains. With the addition of the CNT/CCB hybrid fillers, electron bridges were built inside the self-healing BIIR matrix and changed the BIIR insulator to the conductive BIIR composites. In addition, the incorporation of IL inside the composites served multiple functions: (i) IL covered the BIIR-IM molecular chains and supported the chains’ movement to initiate self-healing, (ii) IL wetted the CNT/CCB surfaces and acted as the electron pathways for connecting end-to-end of the hybrid filler, and (iii) the liquid phases of IL improved the dispersion and distribution of the CNT/CCB inside the BIIR matrix to increase the composites’ properties.

In contrast, when considering the addition of the modified agents IM, IL, and hybrid fillers at 1 and 3 phr, it was found that a wide and non-adherent mark was comparable to pure BIIR. However, it showed better tensile properties and healing efficiency than unmodified BIIR. Furthermore, the modified IM, IL, and hybrid fillers at more than 3 phr were observed to have the most tightly marked area indicated by the yellow line. This confirms the self-healing ability of modified BIIR composites. The results showed that mechanical properties had increased up to the IM:IL:CNT/CCB ratio of 1:1:1/1.5, according to the optimal concentration of 5:5:5/7.5 phr.

### 4.2. Abrasion Ability 

Figure 5 shows the Taber wear resistance of composites using the wear index, where a low wear index indicates excellent wear resistance. It is evident that the modified BIIR with IM, IL, and hybrid fillers has the lowest wear index compared to pure BIIR. The self-healing properties of composite materials are evident, providing effective protection against wear. This can be attributed to the bound rubber between BIIR and fillers, as well as the chain crosslinking of BIIR with the modified agent on the surface. As a result, it reduces the hindrance of molecular chain movement and significantly increases the resistance to deformation, leading to higher wear resistance.

### 4.3. Thermo-Mechanical Properties

Thermo-mechanical properties of the self-healing composites can be examined using temperature scanning stress relaxation (TSSR), which was carried out to assess the network structure and stability under constant stretching. In TSSR studies, stress relaxation is performed under both equivalent and non-peak conditions. The theory of isothermal relaxation of polymers is mainly based on the generalized Maxwell model, as seen in Figure 6. The relaxation modulus (E) of pure BIIR was found to be almost unchanged compared to the modified agent IM, IL, and hybrid fillers of BIIR, which showed a rapid rise in the range from the initial temperature up to approximately 40 °C, as seen in Figure 6A. This is the result of the rubber molecules being subjected to entropy as the temperature increases [17]. The above values then decrease with increasing temperature, corresponding to the detachment of the rubber molecular chain physisorption that is absorbed on the filler surface, which usually occurs at low temperatures up to 120 °C. Furthermore, E was found to increase with the increasing amount of modifiers and fillers. The optimal values were observed at ratios of modified agent and hybrid filler at 5 and 7 phr, which may be due to the higher amount of bound rubber formed by increasing the amount of reinforcement, resulting in increased effectiveness of rubber reinforcement. When considering the relaxation spectra (H) and peak areas in the range from 40 °C to 120 °C, it is used to indicate the amount of bound rubber in the BIIR matrix, as shown in Figure 5B and Table 2, respectively. Figure 6B shows the H, and it was found that peak areas of the pure BIIR are very low, indicating low or almost no bound rubber, which is the proper material without filler [13]. For the modified agent IM, IL, and hybrid fillers of BIIR, the peak areas tend to increase. This is due to an increase in the bound rubber caused by an increase in the modified agent and hybrid fillers.

### 4.4. Dynamic Mechanical Properties

Figure 7 shows the storage modulus (E’) and tan δ as functions of temperature. The glass transition temperature (T_g_) and tan δ at 0 and 60 °C (tan δ_0_ and tan δ_60_) are summarized in Table 3. T_g_ tends to reduce with increasing the modified agent and hybrid fillers. The lower T_g_ observed might be attributed to the adsorption of fillers on the surface of the rubber matrix and crosslinking of the matrix phase in the composites using sulfur and modified agents, which hinders the arrangement of molecular chains. Additionally, the value of Tan δ_max_ indicates the degree of chain mobility and damping properties, as seen in Table 3. Tan δ_max_ is noticeable in pure BIIR, and the value decreases significantly after the addition of the modified agent and fillers. This is attributed to the crosslinking of the rubber matrix in the composites using crosslinking agents and modified agents, limiting the mobility of the rubber molecular chain and allowing it to store energy well so that it can recover itself when exposed to external forces.

When considering tan δ_0_ and tan δ_60_ obtained in this study, as shown in Figure 7, tan δ_60_ tends to increase after the addition of the modified agent and fillers. A high tan δ_60_ refers to low elasticity [18,19]. In contrast, the opposite results were observed for tan δ_0_, which tends to decrease due to the improved filler-rubber interaction resulting from the well-dispersed IM, IL, and hybrid filler in the BIIR matrices. 

### 4.5. Thermal Stability 

Figure 8 relates to the relationship between percent degradation and the temperature of pure BIIR and the modified agent of IM, IL, and hybrid fillers in BIIR composites. Additionally, Table 4 exhibits the thermal-oxidative degradation steps of each composite, i.e., under Nitrogen (N2) conditions to analyze the degradation of the BIIR matrix and under oxygen (O2) conditions to analyze the degradation of C-C bonding in the fillers. It is observed that thermal degradation under N2 conditions decreases with the increasing amount of modified agents and fillers. In the case of thermal degradation under O2 conditions, the degradation increases with the degradation of C-C bonding in the structure of the filler. For BIIR and modified BIIR with IM, IL, and hybrid fillers, it was found that the highest T0 is approximately 363 °C for pure BIIR. Then, T0 decreases as the amount of modified IM, IL, and hybrid fillers increases due to the thermal conductivity of fillers and their effective presence around the BIIR matrix, allowing heat to be distributed inside the material more rapidly. Therefore, increasing the modified agent and hybrid fillers in BIIR results in a decrease in T0. However, when TE is considered, the above values are found to increase with the increasing amount of modified agents and hybrid fillers. This might be attributed to the bound rubber between rubber and filler, as well as increased crosslinking of BIIR and the modified agent. Additionally, this results in obstructing the movement of the chains and hindering the decomposition of chains from heat. Therefore, it requires a higher temperature to break the intramolecular chains of the rubber matrix, resulting in a higher TE.

### 4.6. Factors to Healing Efficiency of Composites

The results showed that mechanical properties increased until the IM:IL:CNT/CCB ratio of 1:1:1/1.5, according to the optimal concentration of 5:5:5/7.5 phr. The healing procedure condition was examined via the application of pressure, temperature, and time. In Figure 9A, it was found that increasing the pressure from 100 psi to 1000 psi significantly increased the healing efficiency. However, when the pressure was further increased to 2000 and 2500 psi, the healing efficiency did not change significantly. This may be the result of increased pressure, which allows the rubber matrix to move closer together, facilitating easier surface bonding due to external pressure. However, when the pressure reaches a certain level, the healing efficiency does not increase significantly because the external pressure does not directly affect the ionic functional group, which is the main mechanism of self-healing. Instead, it only aids in facilitating easier movement of the rubber matrix. In Figure 9B,C, it was found that healing efficiency increased with time, suggesting that longer durations allow the ionic functional groups more time to act. Additionally, higher temperature enables the ionic functional groups to respond more quickly, leading to higher healing efficiency due to the attractive forces between positive and negative charges. As a result of the N-alkylation, the rubber is able to heal. 

## 5. Conclusions

The integration of IM, IL, and the CNT/CCB hybrid filler into BIIR composites marks a groundbreaking advancement in self-healing materials, enhancing intrinsic properties and the reparability of the resulting composites. The presence of IL and IM fosters heightened physical and chemical interactions within the composites. The study’s outcomes showcased an increase in mechanical properties up to the IM:IL:CNT/CCB ratio of 1:1:1/1.5, aligning with the optimal concentration of 5:5:5/7.5 phr. An increase in the IM:IL:CNT/CCB ratios has a tendency to enhance the properties of the composites, both before and after healing. This improvement extends to mechanical properties, self-healing capabilities, as well as thermal and thermo-oxidative stabilities. In this scenario, CNT plays a pivotal role as the primary filler network, forming within the bulk rubber matrix, while CCB acts as the connecting link between cylindrical CNT structures, facilitating electron movement and tunneling within the composites. Additionally, IM is employed to modify BIIR molecular chains, inducing ionic crosslinking and enabling self-healing reactions when surface breakages occur. Furthermore, IL interacts physically with BIIR-IM molecular chains, supporting chain mobility during the de-crosslinking process under controlled conditions. It is worth noting that IL, being in the liquid phase within the composites, assists in the dispersion and distribution of CNT/CCB during the mixing operation processes. The TSSR unveiled thermo-oxidative degradation, where the composites displayed resistance to decomposition at higher temperatures, attributable to a well-formulated BIIR composite composition. In addition, the healing procedure condition was examined via applying pressure, temperature, and time. The composites exhibited robust elasticity at both 0 and 60 °C, demonstrating swift re-crosslinking at suitable pressures and temperatures. This promising formulation holds potential for application in the tire tread and inner liner of commercial car tires, with the goal of utilizing its findings for practical tire manufacturing.

## Figures and Tables

**Figure 1 polymers-15-04023-f001:**
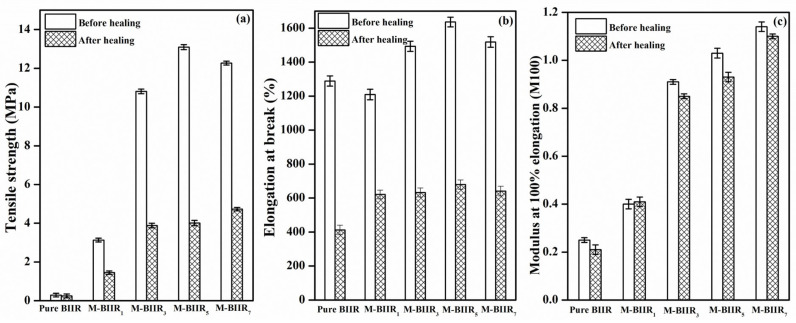
Tensile strength (**a**), elongation at break (**b**), and modulus at 100% elongation (M100) (**c**) of pure BIIR and modified agent of IM, IL, and hybrid fillers of BIIR composites.

**Figure 2 polymers-15-04023-f002:**
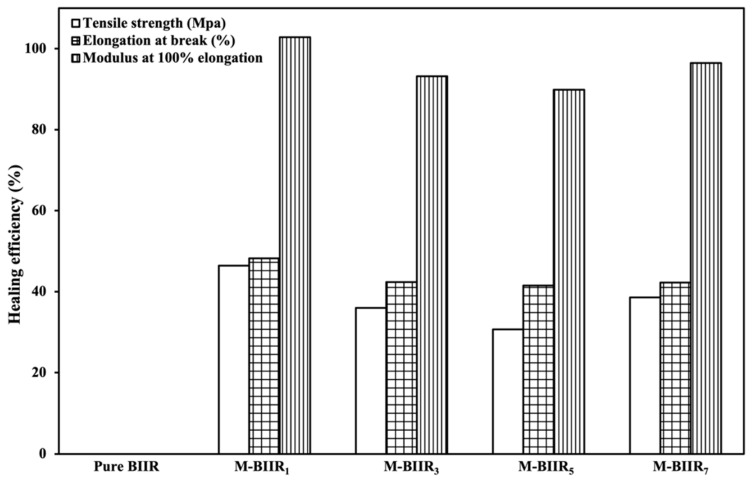
Healing efficiency of BIIR with and without the modified agent of IM/IL and hybrid fillers.

**Figure 3 polymers-15-04023-f003:**
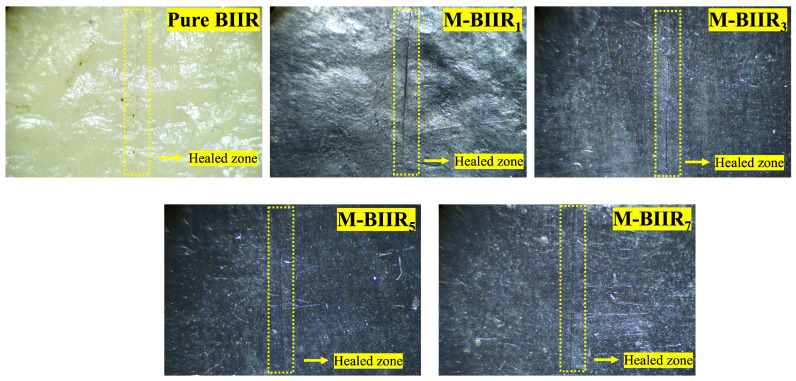
Morphologies of pure BIIR and modified BIIR with IM, IL, and hybrid fillers composites at 0, 3, 5, and 7 phr after healing propagation.

**Figure 4 polymers-15-04023-f004:**
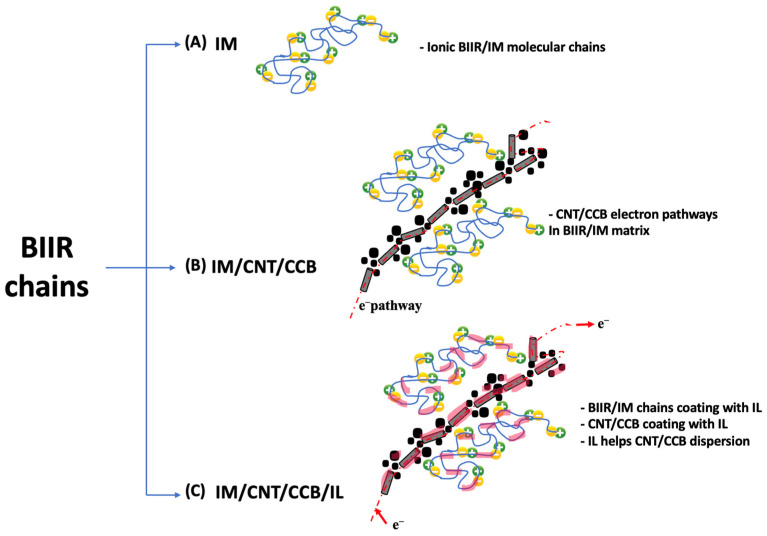
The proposed model of the changes in molecular chain levels of the BIIR before and after the addition of IM (**A**), CNT/CCB (**B**), and IL (**C**).

**Figure 5 polymers-15-04023-f005:**
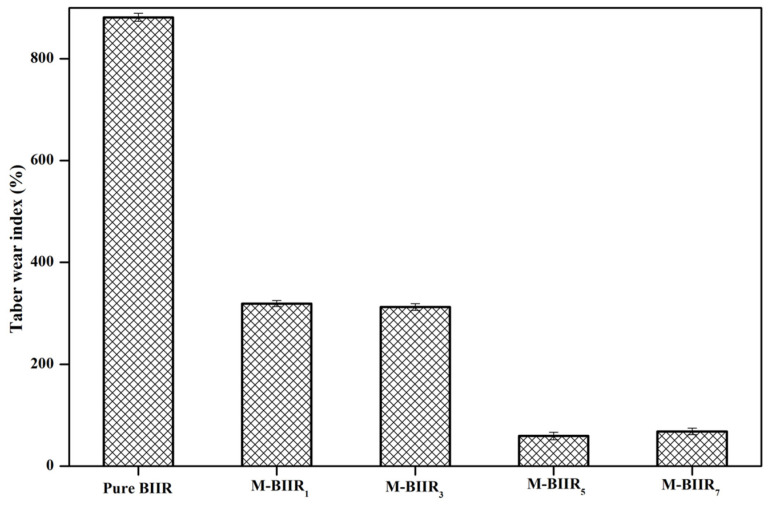
Taber Wear index of pure BIIR and modified agent of IM, IL and hybrid fillers of BIIR composites.

**Figure 6 polymers-15-04023-f006:**
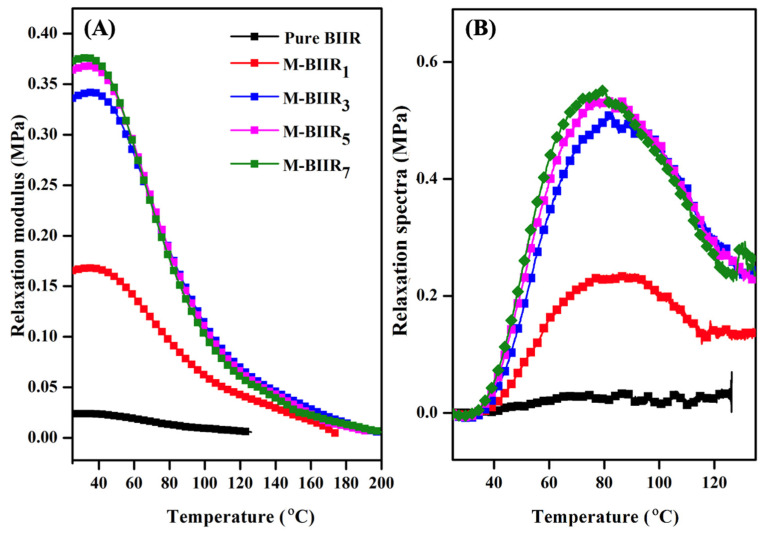
Relationship of Relaxation modulus (**A**) and Relaxation spectra (H) (**B**) of pure BIIR and modified agent of IM, IL, and hybrid fillers of BIIR composites.

**Figure 7 polymers-15-04023-f007:**
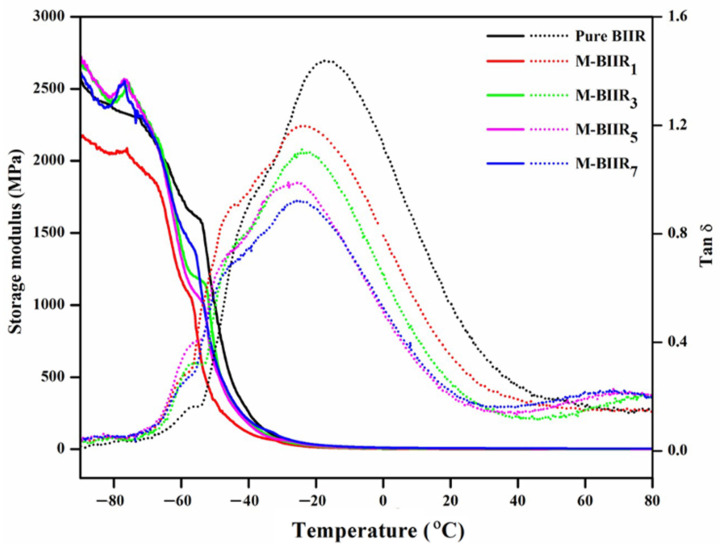
Relationship of the storage modulus and Tan δ with a temperature of pure BIIR and modified agent of IM, IL, and hybrid fillers of BIIR composites.

**Figure 8 polymers-15-04023-f008:**
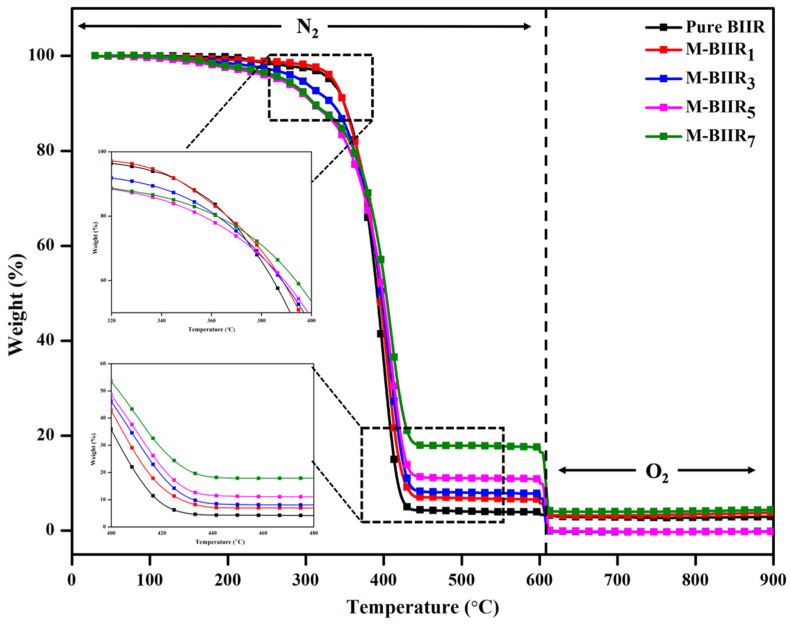
Weight as a function of temperature of pure BIIR and modified agent of IM, IL, and hybrid fillers of BIIR composites.

**Figure 9 polymers-15-04023-f009:**
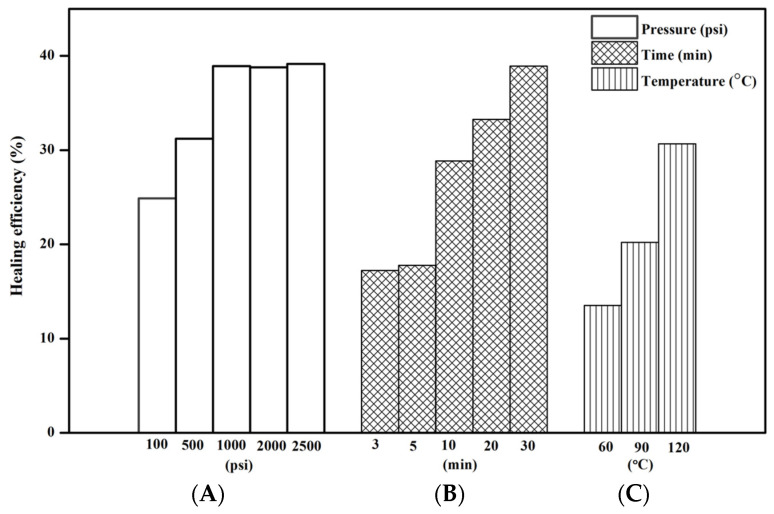
Relationship to factor of healing efficiency (**A**) pressure, (**B**) Time, and (**C**) Temperature of pure BIIR and modified agent of IM, IL, and hybrid fillers of BIIR composites.

**Table 1 polymers-15-04023-t001:** Formulation of pure BIIR and their composites.

Samples	IM	CNT	CB	IL
	(phr)	(phr)	(phr)	(phr)
Pure BIIR	0.0	0.0	0.0	0.0
M-BIIR_1_	1.0	1.0	1.5	1.5
M-BIIR_3_	3.0	3.0	4.5	4.5
M-BIIR_5_	5.0	5.0	7.5	7.5
M-BIIR_7_	7.0	7.0	10.5	10.5

**Table 2 polymers-15-04023-t002:** Peak areas of relaxation spectra (H) in the range from 40 °C to 120 °C of pure BIIR and modified agent of IM, IL, and hybrid fillers of BIIR composites.

Samples	Peak Areas
	(MPa.K)
Pure BIIR	1.75
M-BIIR_1_	13.51
M-BIIR_3_	27.47
M-BIIR_5_	31.85
M-BIIR_7_	32.03

**Table 3 polymers-15-04023-t003:** Dynamic mechanical of pure BIIR and modified agent of IM, IL, and hybrid fillers of BIIR composites.

Sample	T_g_ (°C)	Tanδ _max_	Tanδ _0_	Tanδ _60_
Pure BIIR	−17.20	1.44	1.15	0.16
M-BIIR_1_	−23.96	1.12	0.79	0.15
M-BIIR_3_	−24.13	1.11	0.65	0.15
M-BIIR_5_	−28.16	0.99	0.53	0.19
M-BIIR_7_	−26.10	0.93	0.52	0.20

**Table 4 polymers-15-04023-t004:** Decomposition of two-step and Decomposition temperature of pure BIIR and modified agent of IM, IL, and hybrid fillers of BIIR composites.

Samples	Decomposition (%)		T_DR_ (°C)	
	Step 1 (N_2_)	End Step 1 (O_2_)	* T_0_	* T_E_
Pure BIIR	93.9	4.1	362.1	429.2
M-BIIR_1_	91.5	7.0	363.4	432.6
M-BIIR_3_	89.8	8.1	359.9	437.9
M-BIIR_5_	86.5	11.1	358.8	437.8
M-BIIR_7_	81.4	17.9	358.9	440.0

* T_0_ and T_E_ are initial degradation and final degradation in step 1, respectively.

## Data Availability

Not applicable.

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
