# Peer review of "Effects of Modifying Agent and Conductive Hybrid Filler on Butyl Rubber Properties: Mechanical, Thermo-Mechanical, Dynamical and Re-Crosslinking Properties"

_polymers, 2023, doi:10.3390/polym15194023_

Round 1
Reviewer 1 Report
The bromobutyl rubber composites with butylimidazole, ionic liquid, carbon nanotubes and conductive carbon black were studied in the paper. The results showed that the mechanical properties, healability, thermo-oxidative degradation and other properties of the composites were enhanced by adding the additional fillers.
The following questions or suggestions are for reference only.
The experimental results of five samples are difficult to support the whole conclusion, and more data should be provided.
What are the effects of four fillers on the properties of composites respectively?
How does the change of their dosage affect the properties of composites respectively?
Analyze the process of self-healing of composites from the perspective of chemical structure.
The legend of After healed in Figure 1(b) needs to be corrected.
Minor editing of English language required.
Author Response
Reviewer 1
The bromobutyl rubber composites with butylimidazole, ionic liquid, carbon nanotubes and conductive carbon black were studied in the paper. The results showed that the mechanical properties, healability, thermo-oxidative degradation and other properties of the composites were enhanced by adding the additional fillers.
The following questions or suggestions are for reference only.
- The experimental results of five samples are difficult to support the whole conclusion, and more data should be provided.
ANS According to our best understanding, 5 specimens for each formulation testing can be accepted. However, for avoiding the mistaking, we have a sentence to indicate that the 5 specimens were applied to all the mechanical and abrasion testing as can be indicated in “BLUE” letters in the “Characterization” section.
- What are the effects of four fillers on the properties of composites respectively?
ANS Thank you very much the reviewer for the comments. This is the important point and therefore we added the roles of each chemical in the “Conclusion” section in “BLUE” letters as can be seen below:
“In this scenario, CNT plays a pivotal role as the primary filler network, forming within the bulk rubber matrix, while CCB acts as the connecting link between cylindrical CNT structures, facilitating electron movement and tunneling within the composites. Additionally, IM is employed to modify BIIR molecular chains, inducing ionic crosslinking and enabling self-healing reactions when surface breakages occur. Furthermore, IL interacts physically with BIIR-IM molecular chains, supporting chain mobility during the decrosslinking process under controlled conditions. It is worth noting that IL, being in liquid phase within the composites, assists in the dispersion and distribution of CNT/CCB during the mixing operation processes.”
- How does the change of their dosage affect the properties of composites respectively?
ANS In this case, change the dosage means the change of the IM:IL:CNT/CCB concentration ratios. Regarding the results, the increases of the IM:IL:CNT/CCB tends to increase the properties of the composites before and after healing in terms of mechanical properties, self-healing ability together with thermal and thermos-oxidative stabilities. It is seen that the IM of 5 phr is the suitable for the IL:CNT/CCB ration which made each chemical performing well on their roles by means of BIIR modification, filler dispersion and re-crosslink ability. Nevertheless, in order to avoid misleading, we added the explanation in the “Conclusion” section indicating in “BLUE” letters as described below:
“An increase in the IM:IL:CNT/CCB ratios has a tendency to enhance the properties of the composites, both before and after healing. This improvement extends to mechanical properties, self-healing capabilities, as well as thermal and thermos-oxidative stabilities”
- Analyze the process of self-healing of composites from the perspective of chemical structure.
ANS Thank you very much the reviewer for the suggestion. Regarding to the explanation of the self-healing, we added the proposed model and its interpretation inside the manuscript in the “Results and discussion” section indicating in “BLUE” letters as seen below:
“Therefore, the molecular chains changing after the addition of the proper IM:IL:CNT/CCB ratios can be summarized in the proposed model of Figure 4 for the BIIR composites before and after modification by means of the molecular chains’ changes. It is the seen that the modification of BIIR molecules with IM changes the BIIR chains to be the ionic chains containing of cationic and anionic charges, generating the re-crosslinkable chains. With the addition of the CNT/CCB hybrid fillers, the electron bridges were built inside the self-healing BIIR matrix and change the BIIR insulator to be the conductive BIIR composites. In addition, the incorporation of the IL inside the composites acted as the multi-functional roles; (i) IL had covered on the BIIR-IM molecular chains and supported the chains movement for originating the self-healing, (ii) IL had wetted on the CNT/CCB surfaces and performed as the electron pathways for connecting end-to-end of the hybrid filler, and (III) the liquid phases of IL can improve the dispersion and distribution of the CNT/CCB inside the BIIR matrix for increasing the composites properties.”
- The legend of After healed in Figure 1(b) needs to be corrected.
ANS We have revised the legends to be “Before healing” and “After healing”

Reviewer 2 Report
Interesting paper. This paper shows the results that the incorporation of IM, IL, and the CNT/CCB hybrid filler into BIIR composites represents a groundbreaking leap in the realm of self-healing materials. This advancement bolsters the inherent properties of the resulting composites while also enhancing their reparability.The paper presents the optimal conbination ratio and concentration. Material prperties are discussed.
Some comments:
1. For some figures, take Figure.2 as example, the legend is not clear which indicates which. Please adjust the legend. The plot should be clear to reader.
2. For equation 3, please explain what E and H refer to. The nomenclature should be clear.
3. The font for Figure 1 and 5 are too small.
4. DMA test is conducted at 10Hz, which is very high. Is there any specifc reason? Is it based on real world application?
5. There is typo of the caption of table 3.
Author Response
Reviewer 2
Interesting paper. This paper shows the results that the incorporation of IM, IL, and the CNT/CCB hybrid filler into BIIR composites represents a groundbreaking leap in the realm of self-healing materials. This advancement bolsters the inherent properties of the resulting composites while also enhancing their reparability.The paper presents the optimal conbination ratio and concentration. Material prperties are discussed.
Some comments:
- For some figures, take Figure.2 as example, the legend is not clear which indicates which. Please adjust the legend. The plot should be clear to reader.
ANS Thank you very much the reviewer for your comment. We have checked and revised all the unclear figures to be the better version as can be seen in the revised version of the manuscript.
- For equation 3, please explain what E and H refer to. The nomenclature should be clear.
ANS The assignment of the E and H factors have been added in the manuscript indicating in “BLUE” letters of “Characterization” section.
- The font for Figure 1 and 5 are too small.
ANS The requirement has already revised following the suggestion.
- DMA test is conducted at 10Hz, which is very high. Is there any specifc reason? Is it based on real world application?
ANS Thank you the reviewer for the effective comment. We have realized that we used only 1 Hz under the tests, we revised the typo.
- There is typo of the caption of table 3.
ANS The requirement has already revised following the suggestion.
